

# Detection of syrup adulterants in manuka and jarrah honey using HPTLC-multivariate data analysis

Md Khairul Islam[1,2], Kevin Vinsen[3], Tomislav Sostaric[1], Lee Yong Lim[1] and Cornelia Locher[1,2]

[1] Division of Pharmacy, School of Allied Health, University of Western Australia, Crawley, WA, Australia
[2] Cooperative Research Centre for Honey Bee Products Limited (CRC HBP), Perth, WA, Australia
[3] International Centre for Radio Astronomy Research (ICRAR), University of Western Australia, Crawley, WA, Australia

## ABSTRACT

High-Performance Thin-Layer Chromatography (HPTLC) was used in a chemometric investigation of the derived sugar and organic extract profiles of two different honeys (Manuka and Jarrah) with adulterants. Each honey was adulterated with one of six different sugar syrups (rice, corn, golden, treacle, glucose and maple syrups) in five different concentrations (10%, 20%, 30%, 40%, and 50% w/w). The chemometric analysis was based on the combined sugar and organic extract profiles' datasets. To obtain the respective sugar profiles, the amount of fructose, glucose, maltose, and sucrose present in the honey was quantified and for the organic extract profile, the honey's dichloromethane extract was investigated at 254 and 366 nm, as well as at T (Transmittance) white light and at 366 nm after derivatisation. The presence of sugar syrups, even at a concentration of only 10%, significantly influenced the honeys' sugar and organic extract profiles and multivariate data analysis of these profiles, in particular cluster analysis (CA), principal component analysis (PCA), principal component regression (PCR), partial least-squares regression (PLSR) and Machine Learning using an artificial neural network (ANN), were able to detect post-harvest syrup adulterations and to discriminate between neat and adulterated honey samples. Cluster analysis and principal component analysis, for instance, could easily differentiate between neat and adulterated honeys through the use of CA or PCA plots. In particular the presence of excess amounts of maltose and sucrose allowed for the detection of sugar adulterants and adulterated honeys by HPTLC-multivariate data analysis. Partial least-squares regression and artificial neural networking were employed, with augmented datasets, to develop optimal calibration for the adulterated honeys and to predict those accurately, which suggests a good predictive capacity of the developed model.

Corresponding author
Cornelia Locher,
connie.locher@uwa.edu.au

## INTRODUCTION

Honey has been regarded as a nutritional and medicinal food for thousands of years (*Kuropatnicki, Kłósek & Kucharzewski, 2018*; *Vorlová, 2015*). About 70–80% of the total solid content of honey is made of sugars (*Pita-Calvo & Vázquez, 2018*), in particular fructose, glucose, maltose and sucrose (*Arvanitoyannis et al., 2005*). Sugars play a vital role, not only in providing crucial nutrition to bee colonies, but also for maintaining the osmolality of the honey (*Olaitan, Adeleke & Ola, 2007*) and thus its self-preservation. As the bulk of the honey is made of sugars, sugar adulterations are common (*Amiry, Esmaiili & Alizadeh, 2017*), either by adding sugar syrups to the honey matrix to increase its bulk volume or by feeding bees with sugar syrups to increase honey production (*Du et al., 2015*; *Rios-Corripio, Rojas-López & Delgado-Macuil, 2012*). Glucose syrup, maple syrup, rice syrup, brown rice syrup, treacle syrup, golden syrup and high fructose corn syrups are commonly used in post-harvest honey adulteration (*Abdel-Aal, Ziena & Youssef, 1993*; *Amiry, Esmaiili & Alizadeh, 2017*; *Chen et al., 2014*; *Ferreiro-González et al., 2018*).

A wide range of methods exist for the detection of adulterants in honey, including $^{13}C/^{12}C$ Stable Carbon Isotope Ratio Analysis (*Bertelli et al., 2010*; *Jamróz et al., 2014*), High-Performance Liquid Chromatography-Mass Spectrometry (HPLC-MS) (*Wen et al., 2017*), Gas Chromatography–Mass Spectrometry (GC-MS) (*Tahir et al., 2016*), Nuclear Magnetic Resonance (NMR) Spectrometry (*Donarski, Jones & Charlton, 2008*; *He et al., 2020*; *Lolli et al., 2008*; *Spiteri et al., 2015*), Near-infrared (NIR) spectroscopy (*Başar & Özdemir, 2018*; *Mishra et al., 2010*), Raman spectroscopy (*Corvucci et al., 2015*), Laser-Induced Breakdown Spectroscopy (*Nespeca et al., 2020*; *Zhao et al., 2020*), Fourier Transform Infrared (FT-IR) Spectroscopy (*Hennessy, Downey & O'Donnell, 2010*; *Riswahyuli et al., 2020*; *Wang et al., 2010*), Plasma Mass Spectrometry (*Voica, Iordache & Ionete, 2020*) and X-ray fluorescence (*Fiamegos et al., 2020*). Chemometric analyses to confirm the geographic origin and authenticity of honey are also commonly used, based on routine chemical quality parameters (*Sant'Ana et al., 2012*), physicochemical characteristics (*Bentabol Manzanares et al., 2017*; *Rodríguez-Flores et al., 2019*; *Scholz et al., 2020*; *Truzzi et al., 2014*), mineral composition data (*Grembecka & Szefer, 2013*; *Voica, Iordache & Ionete, 2020*), amino acid and protein profiles (*Kečkeš et al., 2013*), phenolic components (*Ciucure & Geană, 2019*; *Dżugan et al., 2018*; *Rodríguez-Flores et al., 2019*; *Tonello et al., 2016*; *Wen et al., 2017*) as well as volatile content evaluation (*Tahir et al., 2016*).

All these methods have their limitations; NMR, GC, plasma mass spectrometry and X-ray fluorescence, for instance, require expensive instrumentation and might thus not be suitable for regular quality control purposes. Furthermore, NMR analysis is based on a spatial reference database and also requires expert personnel for data interpretation (*Ferreiro-González et al., 2018*). $^{13}C/^{12}C$ stable carbon isotope ratio analysis, NIR and FT-IR on the other hand are limited in their analytical capacity to the detection and/or quantification of sugars only (*Se et al., 2019*), which is not always suitable for the discrimination between different honeys. High-Performance Thin-Layer Chromatography

(HPTLC) analysis presents an attractive analytical alternative, in particular for day-to-day quality control purposes, as it is less expensive and can provide rich data (*e.g.*, quantification of sugars and organic extract fingerprint for floral source identification) (*Islam et al., 2020a*; *Islam et al., 2020b*; *Locher, Neumann & Sostaric, 2017*; *Locher et al., 2018*). HPTLC is a sophisticated and increasingly popular tool for the analysis of complex matrices. It facilitates semi-to fully automated analysis, it is very efficient as it supports the parallel analysis of multiple samples in a single run, it provides options for a wide range of chemical derivatisations (untargeted and targeted analysis) and has the ability of hyphenation with other analytical platforms and multivariate data analysis (*Islam et al., 2020b*; *Li et al., 2019*; *Stanek & Jasicka-Misiak, 2018*; *Stanek, Kafarski & Jasicka-Misiak, 2019*). Multivariate Data Analysis (MVDA), including cluster analysis (CA), hierarchical clustering analysis (HCA), principal component analysis (PCA), principal component regression (PCR), partial least-squares regression (PLSR) and artificial neural networks (ANN), have been specifically designed for the analysis and visualisation of complex sets of samples like those presented by honey and honey adulterant mixtures (*Amiry, Esmaiili & Alizadeh, 2017*; *Arvanitoyannis et al., 2005*; *Başar & Özdemir, 2018*; *Bertelli et al., 2010*).

The aim of this study was to determine the feasibility of HPTLC-based profiling in combination with MVDA for the quality control of Manuka and Jarrah honeys, specifically the detection of post-harvest adulterations with a range of sugar syrups. It was found that the combination of HPTLC and multivariate data analysis can not only assist in confirming and quantifying these adulterations, but also in identifying some of the adulterants used.

## MATERIALS AND METHODS

### Chemicals and reagents

All chemicals and reagents were of analytical grade. Commercial sugar syrups and honeys (Table 1) were obtained from supermarkets and other commercial suppliers in Western Australia.

### Sample preparation

Adulterated honey samples were prepared by mixing the respective syrups (rice, corn, golden, treacle, glucose and maple) with honey (Manuka and Jarrah) in a final concentration of 10%, 20%, 30%, 40% and 50% (w/w) respectively. The samples were warmed in a water bath at 36 °C for about 30 min to assist with their mixing into homogenous blends. As described in *Islam et al. (2020b)* in more detail, for sugar analysis one mg/mL samples of the two honeys, six syrups and 60 adulterated honeys were prepared in 50% aqueous methanol. The organic honey extracts were obtained as previously described in *Locher, Neumann & Sostaric (2017)* and *Locher et al. (2018)*. In brief, approximately one g of each sample was mixed with two mL of deionised water and extracted three times with five mL of dichloromethane. After drying the combined organic extracts with anhydrous $MgSO_4$ followed by filtration, the solvent was evaporated at ambient temperature and the resulting extracts stored at 4 °C. Prior to HPTLC analysis

**Table 1 Commercial syrups and honey samples.**

|  | Sample name | Label and packaging information | Reference ID |
|---|---|---|---|
| Honeys | Manuka | Australian Manuka (MGO 514+) Barnes Naturals Pty Ltd. | MAN |
|  | Jarrah | Boyanup Jarrah Sweet As Apiary, WA | JAR |
| Syrups | Rice | Organic Rice Syrup Pureharvest, VIC | RIC |
|  | Corn | Corn Malt Syrup Korea Connections Ptl Ltd. | COR |
|  | Golden | CSR Golden Syrup Sugar Australia Pty Ltd. | GOL |
|  | Treacle | CSR Treacle Syrup Sugar Australia Pty Ltd. | TRE |
|  | Glucose | Queen Glucose Syrup Queen Fine Foods Pty Ltd, QLD | GLU |
|  | Maple | Queen Maple Syrup Dr. Oetker Queen Australia, QLD | MAP |

they were reconstituted in 100 μL of dichloromethane. A solution of 4,5,7-trihydroxyflavanone (0.5 mg/mL) in methanol was prepared as a reference solution.

## Instrumentation and sample analysis

The HPTLC sugar and organic honey extract profiles of all samples were obtained using a CAMAG HPTLC instrumentation (Muttenz, Switzerland) following previously established methods (*Islam et al., 2020b*; *Locher, Neumann & Sostaric, 2017*; *Locher et al., 2018*).

In brief, for sugar analysis, three μL of each sample were applied as eight mm wide bands eight mm from the lower edge and 20 mm from the side edge of the HPTLC plate (20 cm × 10 cm glass-backed silica gel 60 $F_{254}$ plates) using a semi-automated HPTLC application device (Linomat 5, CAMAG, Muttenz, Switzerland). Track distances were 11.4 mm with 15 tracks on each 20 × 10 HPTLC plate, which represents the default plate setting. The development chamber was saturated for 60 min (33% relative humidity) and the plates developed to a migration distance of 85 mm in 1-butanol: 2-propanol: aqueous boric acid (5.0 mg/mL, 3:5:1 *v/v/v*). After drying for 5 min, the plates were derivatised with two ml of aniline-diphenylamine-phosphoric acid reagent (CAMAG Nozzle Spraying Derivatiser, Yellow Nozzle, Level 5) and heated for 10 min at 115 °C (CAMAG TLC Plate Heater III). After cooling to room temperature for two min, the plates were analysed at white light with the HPTLC imaging device (TLC Visualizer 2, CAMAG, Muttenz, Switzerland).

To obtain the organic extract profile, the reference solution (four μL) and the reconstituted extracts (five μL) were applied as eight mm bands at eight mm from the lower edge and 20 mm from the side edge of the HPTLC plates (20 cm × 10 cm glass-backed silica gel 60 $F_{254}$ plates) with a semi-automated HPTLC application device (Linomat 5, CAMAG, Muttenz, Switzerland). Track distances were 11.4 mm with 15 tracks

on each 20 × 10 HPTLC plate, which represents the default plate setting. The development chamber was saturated for 20 min (33% relative humidity) and the plates developed to a final distance of 70 mm in toluene: ethyl acetate: formic acid (6:5:1 *v/v/v*). After drying for 5 min, the chromatographic results were documented at 254 and 366 nm using the HPTLC imaging device (TLC Visualizer 2, CAMAG, Muttenz, Switzerland) before being derivatised with three ml vanillin with sulfuric acid reagent (CAMAG Nozzle Spraying Derivatiser, Yellow Nozzle, Level 3) and being heated for 3 min at 115 °C (CAMAG TLC Plate Heater III). After cooling to room temperature for 2 min, the obtained images were again recorded at white light and 366 nm. The chromatographic analysis was performed using VisionCATS software (Version 2.5, CAMAG, Muttenz, Switzerland), which was also used to control the individual instrumentation modules.

## Data acquisition and chemometric analysis

All multivariate data analysis were performed on a 1,253 × 68 matrix, which included data derived from the HPTLC sugar and phenolic organic honey extract profiles of the two pure honeys (Manuka and Jarrah), the six syrup adulterants (rice, corn, golden, treacle, glucose and maple) and the resulting 60 adulterated honeys. Sugar data included quantities of fructose, glucose, sucrose and maltose. From the organic honey extract HPTLC profiles, four sets of images (R 245 and R 366, and T white and R 366 after derivatisation with vanillin reagent) were converted into their respective chromatograms. The intensity (AU) of bands expressed as absorption peak height were extracted for data calculation along with their corresponding Rf values. To reduce complexity, only bands between Rf 0.05 and Rf 0.60 were considered, as this captured the majority of all detected bands (*Islam et al., 2021*). The obtained data sets were standardised and the multivariate data analysis performed using R and R Studio (Version 1.3.959) (*Team RC, 2020a*; *Team RC, 2020b*), Python3.9 (*Van Rossum & Drake, 2009*) and PyTorch (*Paszke et al., 2019*).

## Multivariate data analysis

Multivariate data analysis included non-supervised techniques, like cluster analysis and principal component analysis, as well as supervised techniques, including principal component regression, partial least-squares regression and machine learning (*Agatonovic-Kustrin & Beresford, 2000*; *Donarski, Jones & Charlton, 2008*; *Se et al., 2019*; *Vorlová, 2015*; *Zhao et al., 2020*).

## Chemometric validation

The sugar contents data had a dynamic range of 0 to ~2,000. All the methods chosen for the analysis prefer standardised data in the number range −1 to 1 or 0 to 1; so, the data was standardised. With the ANN, the AU values of the organic extracts naturally have a range from 0 to 1 and thus did not need standardising. To ensure the supervised models would work with unseen data, and not just repeat the labels of the samples that they had just seen, cross validation and k-fold cross validation was used. This involved holding back 30% of the data as a test set to validate the training of the models. With the basic raw data only 68 samples were available. This meant standard cross validation was

inappropriate for PCR, PLSR and ANN, and k-fold cross validation was required. k-Fold cross validation divides all the samples into *k* groups of samples, called folds of equal sizes (if possible). The prediction function is learned using *k-1* folds, and the fold left out is used for testing. Each fold is used for testing once. This process was applied to supervised PCR, PLSR and ANN chemometric techniques. For PCR and PLSR Root Mean Square Error of Cross Validation (RMSECV), Root Mean Square Error (RMSE) and Root Mean Square Error of Prediction (RMSEP) were deemed useful to predict the validation parameters.

### Data augmentation

This study aimed to predict the honey/syrup type and the adulteration level of all samples, so all 68 classes. The original data set was too small to be used directly as input to train an ANN, PCR, PLSR, and potentially, a PCA. To address this, the fact that an analysis run using HPTLC is not deterministic was used. As different runs will produce small various in AU and the Rf value can slightly drift, this was used to augment the original dataset by repeating the following process 50 times:

- For the bands obtained in four different analysis conditions (R 245 and R 366, and T white and R 366 after derivatisation)-a drift of ±0.0173 Rf and adding Gaussian noise with a standard deviation of 1.25 times the standard deviation of the original AU value. The data was separated into the four light bands and the standard deviation of each band was calculated.
- For the sugar content values–Gaussian noise with a standard deviation of 5% of the value was add to the samples.

This increased the dataset from 68 to 71,468 ($50 \times 21 \times 68$ + the original 68) as shown in Fig. 1 (and Materials-S1) and Fig. 2 (and Materials-S2).

## RESULTS AND DISCUSSIONS

### HPTLC fingerprints and corresponding chromatograms

The obtained HPTLC sugars profiles of honey were very simple as they only captured glucose (Rf 0.30) and fructose (Rf 0.14) within the limit of quantification, as can be seen in Fig. 3. Apart from those two sugars, a very faint signal for maltose (Rf 0.20) was also noted, but it was well below the limit of quantification. Honeys adulterated with corn, rice or glucose syrup contained significant amounts of maltose, which was easily quantifiable, whereas honeys adulterate with golden, treacle and maple syrup were characterised by significant amounts of sucrose, which was also easily quantifiable (*Islam et al., 2020a*). In terms of their respective organic extracts most sugar syrups did not present any major bands. Thus, their incorporation into a honey only decreased the intensity of the major bands present in the organic extract of the pure honey. Only maple syrup extract was characterised by a major band at Rf 0.41 (data not presented), which could not be detected in the two honey extracts and therefore acted as signifying band for this particular syrup. Increasing concentrations of adulterants decreased the intensity of the bands in MAN and JAR accordingly (Figs. 4 and 5; only MAN shown for illustrative purposes).
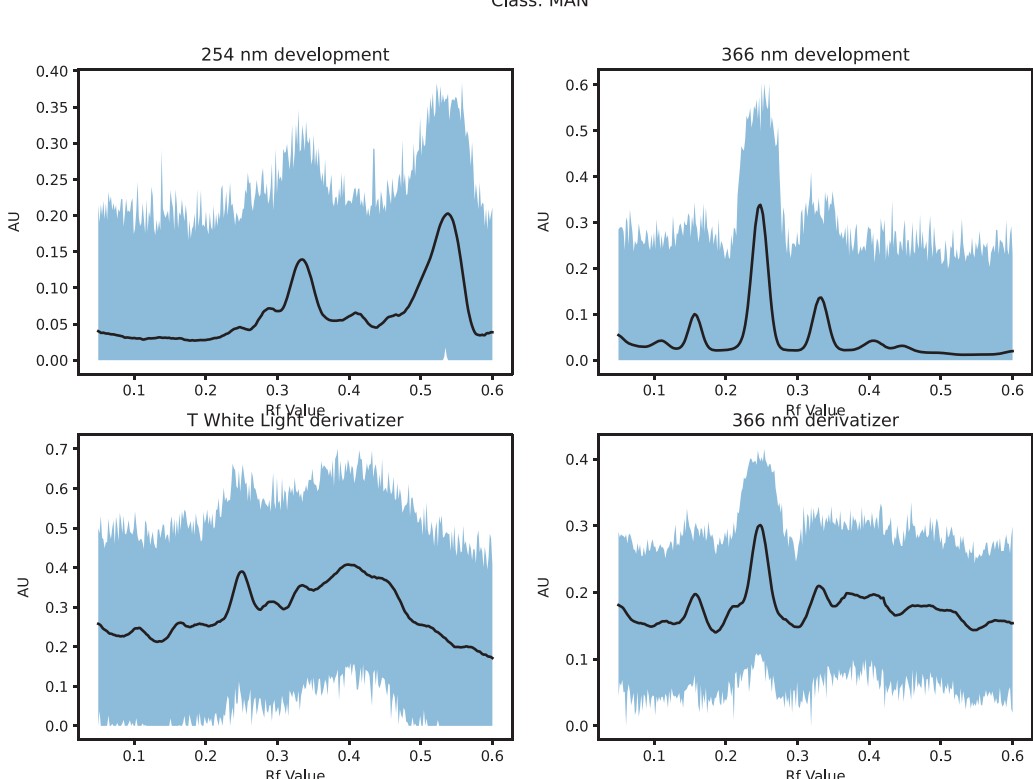

**Figure 1 Augmented dataset for organic extracts Rf *vs* intensity values.**

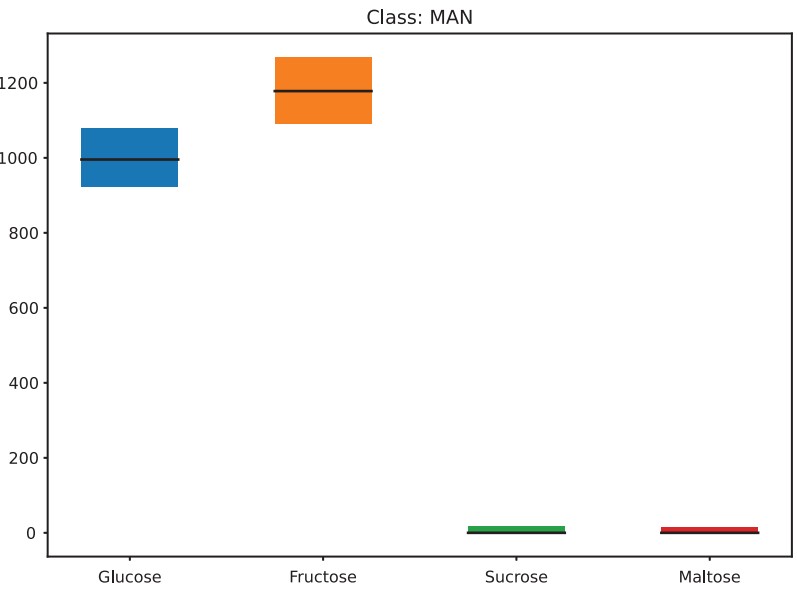

**Figure 2 Augmented dataset for sugar content values.**

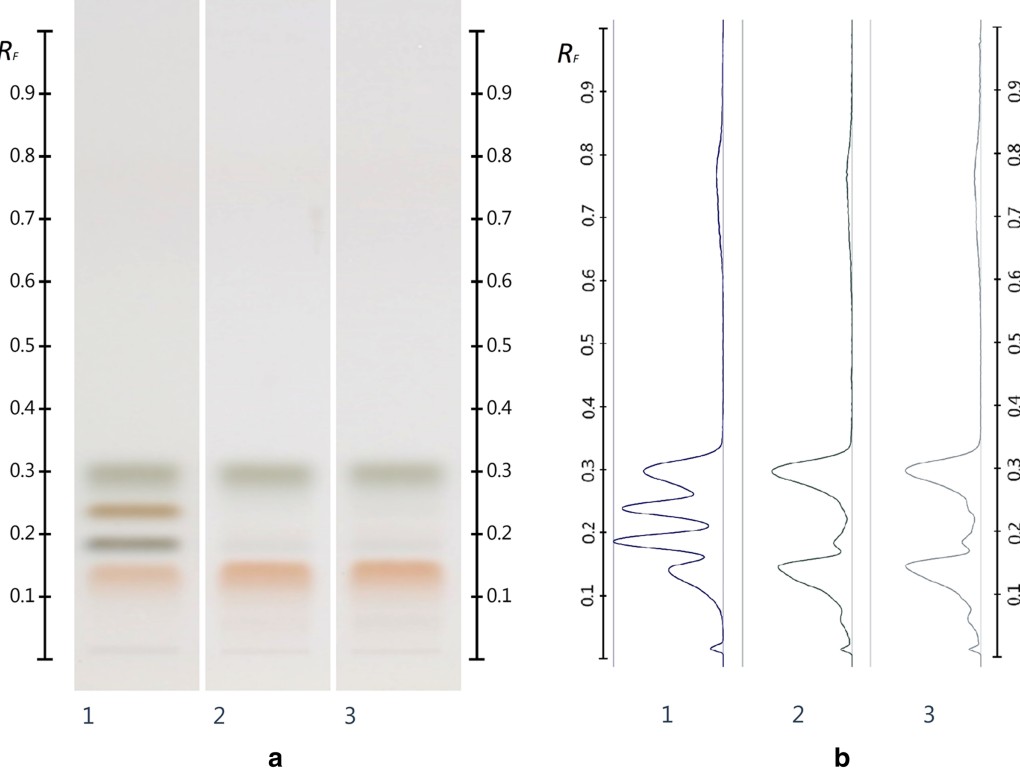

**Figure 3 Images taken at T White light after derivatisation with aniline-diphenylamine-phosphoric acid reagent.** Track 1–Standards (fructose, maltose, sucrose and glucose in increasing Rf values), Track 2–MAN, Track 3–JAR; three μL of each 50% aqueous methanolic solution (A) and their respective chromatograms (B).

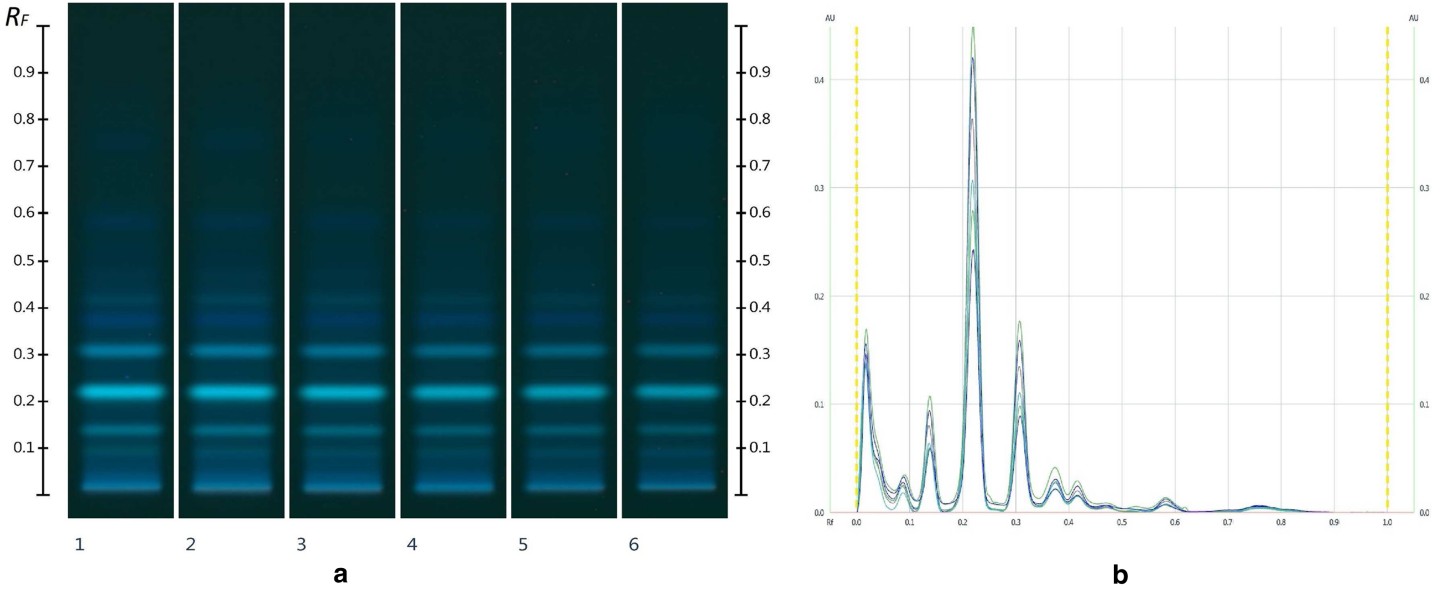

**Figure 4 Images taken at R 366 after development (A) and their respective chromatograms (B).** Track 1–MAN, Track 2–MAN-RIC 10%, Track 3–MAN-RIC 20%, Track 4–MAN-RIC 30%, Track 5–MAN-RIC 40% and Track 6–MAN-RIC 50%; five μL of each extract.
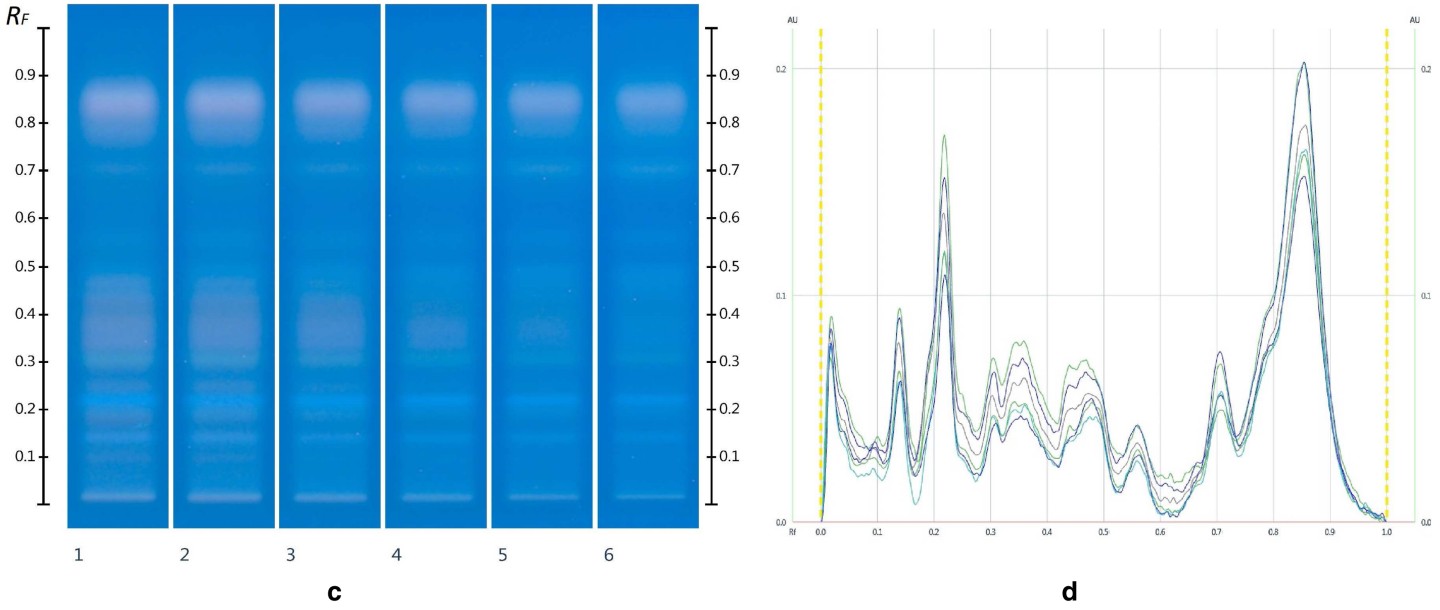

**Figure 5 Images taken at R 366 after derivatisation with vanillin reagent (C) and their respective chromatograms (D).** Track 1–MAN, Track 2–MAN-RIC 10%, Track 3–MAN-RIC 20%, Track 4–MAN-RIC 30%, Track 5–MAN-RIC 40% and Track 6–MAN-RIC 50%; five μL of each extract.

## Cluster analysis

Cluster analysis was applied in this study in order to identify interrelationships between different honey groups. Clustering allows the grouping of similar data points in such a way that points in the same group, known as cluster, are more similar to each other than points in other groups. There are several types of cluster analysis, in this paper Hierarchical Clustering, K-Means Clustering and Density Based Clustering were investigated.

### Hierarchical clustering

Hierarchical clustering (HC), an unsupervised clustering algorithm, is one of the most popular and easy to understand clustering techniques. It starts with placing each observation in its own cluster and then stepwise merging clusters by analysing the distances between adjacent points. HC gives an indication not only to what extent individual points are similar to each other but also how similar different clusters are to each other. The number of significant clusters found and displayed in the respective dendrogram is based on the Euclidian distance of the normalised data (*Dżugan et al., 2018*). The hierarchical clustering (average method) for honeys, syrups and adulterated honeys is shown in Fig. 6.

The dendrogram derived from the study's dataset displays three main clusters. The first contains sugar syrups (with the exception of TRE), the other two contain the pure and adulterated honeys. Within the other clusters two major sub-cluster can be distinguished; one contains pure Manuka and syrup adulterated Manuka honeys, the other contains pure Jarrah and syrup adulterated Jarrah honeys. These findings demonstrate the usefulness of hierarchical clustering in distinguishing syrups from pure honeys and syrup adulterated honeys.
Hierarchical Clustering Dendrogram

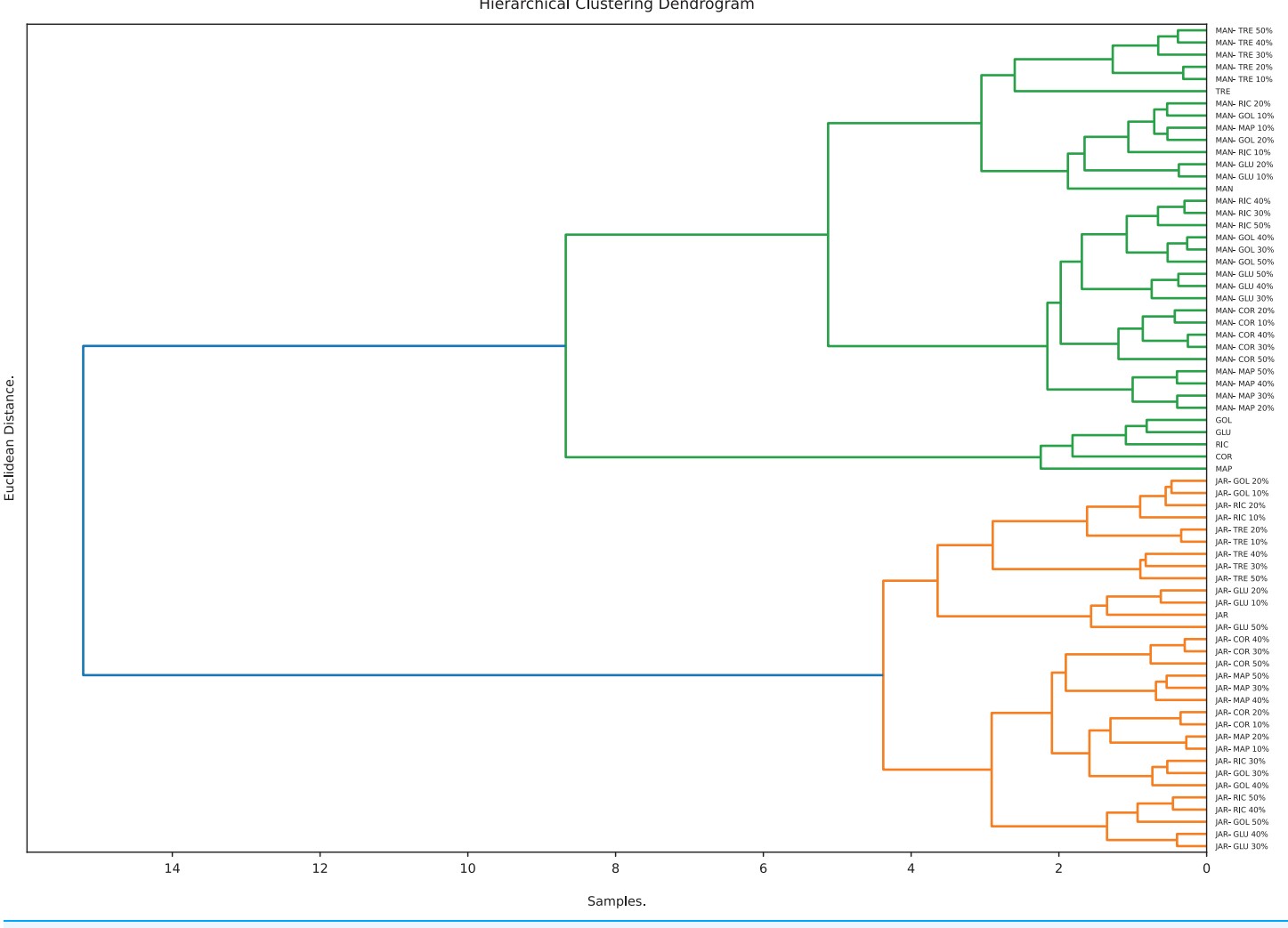

**Figure 6 Hierarchical clustering analysis.**

### K-means clustering

K-means clustering is also a very simple and popular unsupervised algorithm. Typically, unsupervised algorithms use information from datasets as input vectors. A target number k is set first, which refers to the number of centroids needed in the dataset (*Ghosh & Dubey, 2013*). A centroid is the imaginary or real location representing the centre of the cluster (*Fränti, Rezaei & Zhao, 2014*). Every data point is allocated to each of the clusters through reducing the in-cluster sum of squares distance between data points and all centroids. In this analysis we used the elbow method to select the number of clusters k to be four and the clusters sizes (K-means clustering with four clusters) were 22, 5, 26, 15 (between_SS/total_SS = 59.5%) with higher "between Sum of Squares (between_SS)/total Sum of Squares (total_SS)" values signifying better cluster differences values (*Janrao, Mishra & Bharadi, 2019*). The scattered plot of the four clusters of Glucose + Fructose *vs* Sucrose + Maltose are shown in Fig. 7. In the cluster scatter plot, Manuka and Jarrah are clearly separated from the syrup and adulterated honeys (Fig. 7).

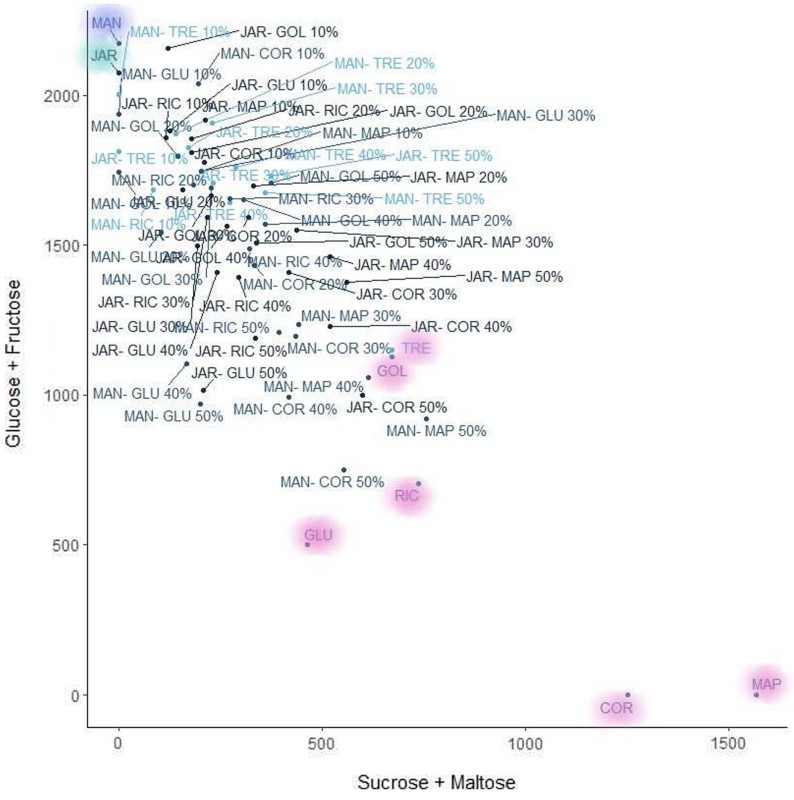

**Figure 7  K-means clustering.**               

### Density based clustering

DBSCAN is a density based clustering algorithm and is based on the concept of grouping samples into clusters if they are connected to one another by density populated areas in order for samples to be clustered into various shapes and sizes, in which the clusters are not sensitive to noise. In this analysis, the eps (*epsilon* or ε-neighbourhood) value, the radius of neighbourhood around a point (*Birant & Kut, 2007*), was set to 21 and minPts as the minimum number of neighbours within the "eps" radius (*Ruiz, Spiliopoulou & Menasalvas, 2007*) was set to three to optimise the clustering results. At eps (21) and minPts (3), the analysis resulted in five clusters (Fig. 8) and seven noise points. As the density-based clusters are not sensitive to noise, the analysis assigns samples to different clusters even within cluster data points and the noise points can be within the clusters but marked as different colours (Fig. 9). The Manuka honey and Manuka adulterated samples were assigned to two different clusters based on their density population and so were Jarrah honey and Jarrah adulterations. The four sugar syrups formed a cluster away from both the Manuka adulterated and Jarrah adulterated clusters. Examination of the seven noise points revealed that Point 1 was Manuka, Point 2 was Jarrah, Point 6 was maple syrup and Point 7 was treacle syrup. Thus, both pure Manuka and Jarrah honey were clearly separated from all other clusters.

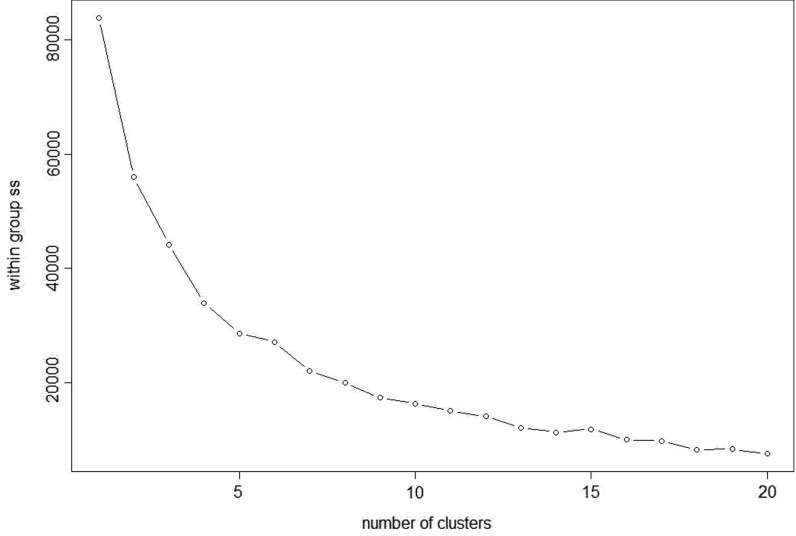

**Figure 8 Density based scree plot.**

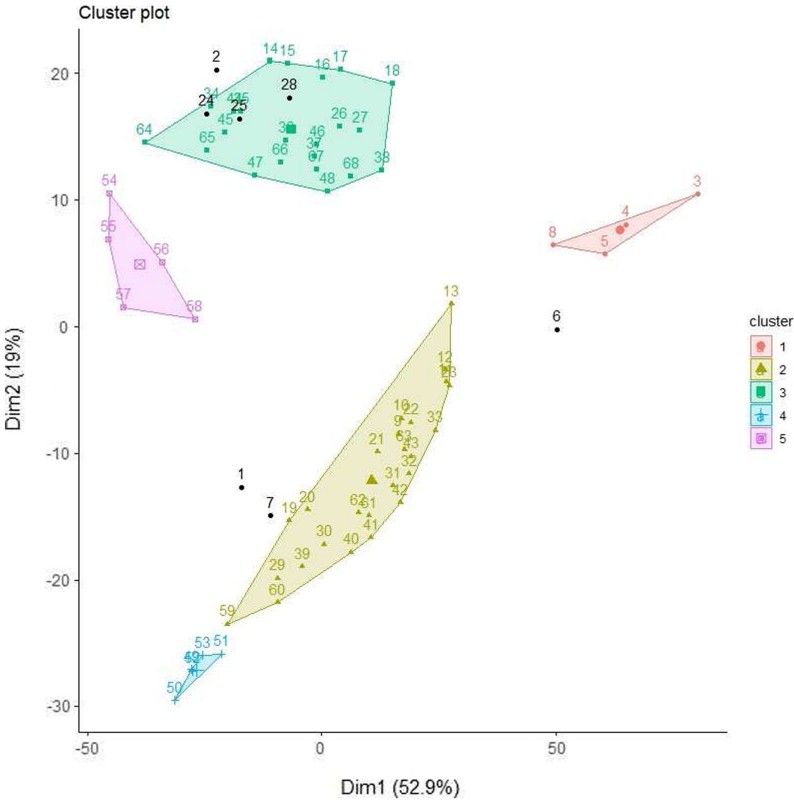

**Figure 9 Density based scattered plot.**

## Principal component analysis

Unsupervised non targeted PCA was conducted to reduce the dimension of the original data to a smaller number of variables or components by examining the relationship between measured parameters. This allowed to explore and model the experimentally

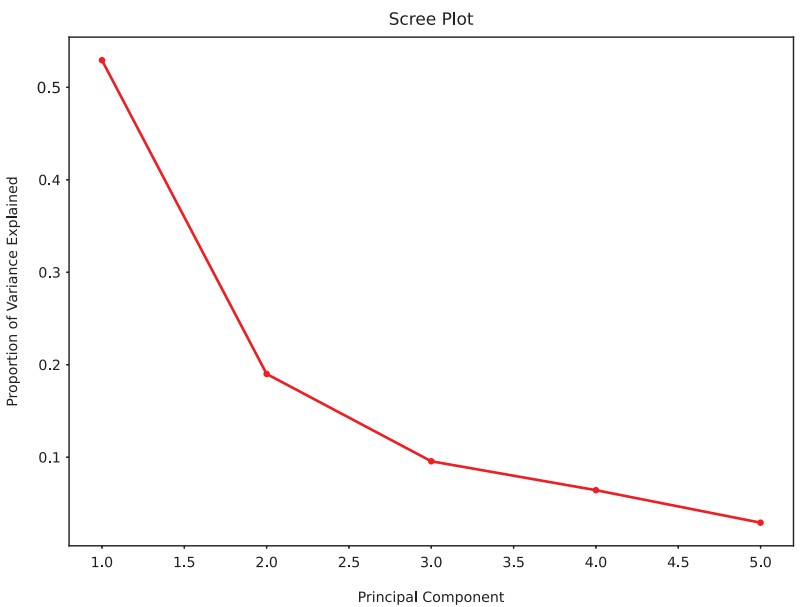

**Figure 10 Principal component analysis summary scree plot.**

derived data, evaluating their correlation and variability; and to represent data visually into a linear transformation (*Hennessy, Downey & O'Donnell, 2010*; *Voica, Iordache & Ionete, 2020*) such as a scree plot and a loading plot.

The PCA scree plot allows to explore the possible number of clusters and variability among the clusters. Principal Component 1 (PC1) and Principal Component 2 (PC2) had the greatest variability, PC2, PC3 and PC4 had marginal variability. Little variability could be detected for PC5 and beyond (Fig. 10).

Although similar chromatographic profiles were obtained for all honey samples (Figs. 3–5), PCA can easily categorise the obtained data into three different clusters based on principal component 1 (PC1) and principal component 2 (PC2) (Fig. 11). The two main PCs, PC1 and PC2, accounted for 52.9% and 19%, thus taken together for 71.9% of total variability.

A clear differentiation between two clusters can be noted. Cluster 1 contains Manuka and Manuka-syrup adulterated honeys and Cluster 2 contains Jarrah and Jarrah-syrup adulterated honeys. Further exploring the PCA data reveals that in Cluster 1 pure Manuka honey is clearly separated from the other data points within this cluster, the same can be noted for Cluster 2, where pure Jarrah honey is also clearly separated from the adulterated samples (Fig. 11). Five of the sugar syrups (except TRE) formed a separate cluster/group. These findings demonstrate that the PCA model can discriminate not only between pure honeys and sugar syrups but also between different syrup adulterated honeys.

When PCA was applied to the augmented dataset, the general pattern of PC1 *vs* PC2 was retained, but the importance of PC1 and PC2 was significantly reduced (Figs. 12 and 13) due to the added noise.

Peer]

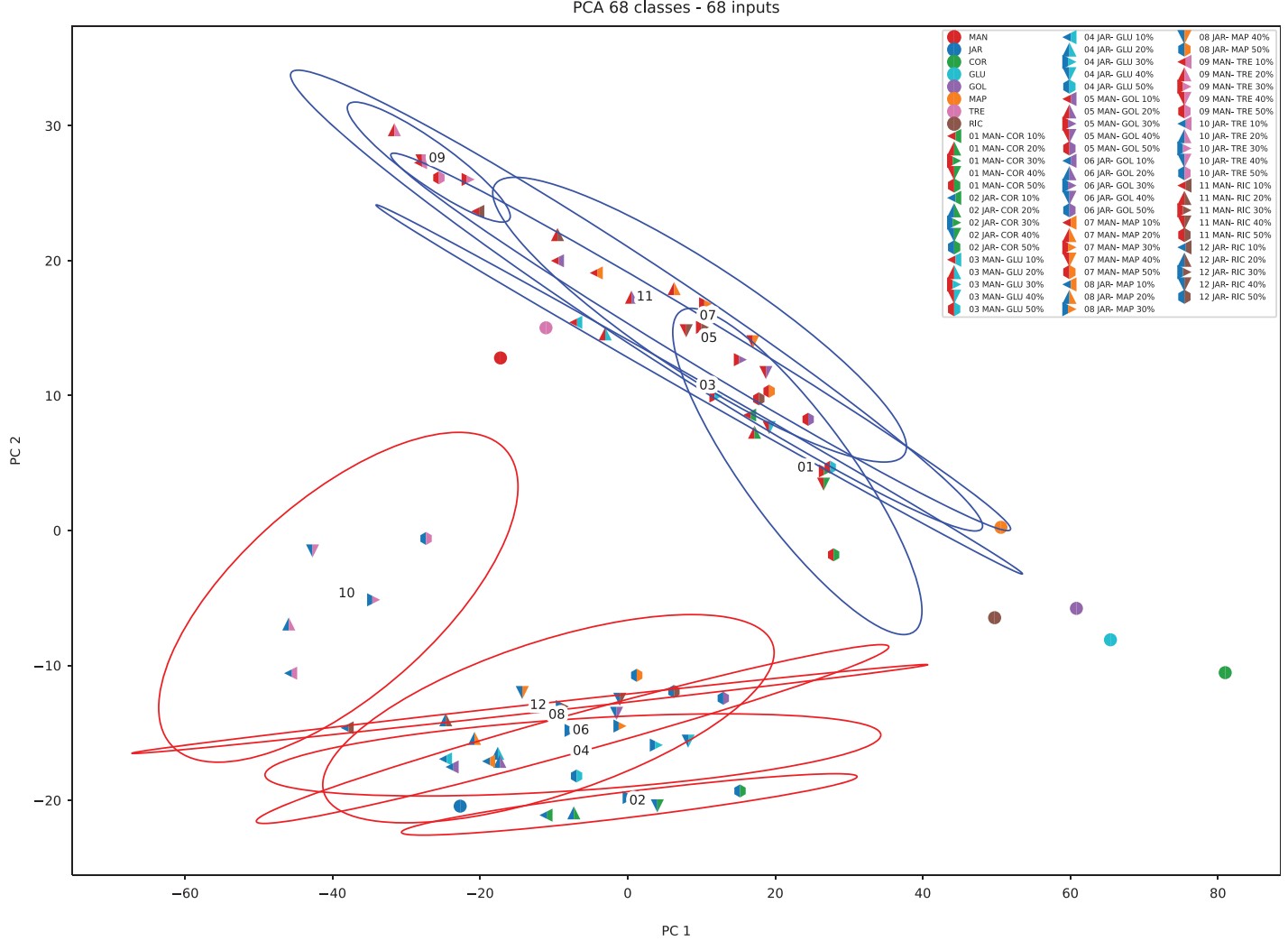

**Figure 11  PCA biplot with clustering.**

## Partial least squares regression and principal component regression

PLS is a classical linear multivariate regression tool, whilst PCR is a regression analysis technique that is based on PCA. These techniques were used to derive the levels of the sugars. The important values are those for Sucrose and Maltose, which if the levels are too high is an indication that the honey is adulterated. For completeness we also derived the Glucose and Fructose values. Supervised PLS and PCR were performed on the standardised data set and the augmented data set to independently calculate the values of the Glucose, Fructose, Sucrose and Maltose content from all the other variables (including the AU values). Generally, the number of PLS factors and the characteristic variables of the data can affect the performance of the PLS model. Too few PLS factors tend to decrease the reliability of the model, too many in turn might increase the model's complexity and weaken its stability (*Ferreiro-González et al., 2018*; *He et al., 2020*). In this work, the number of PCs was optimised by cross-validation in a model calibration process
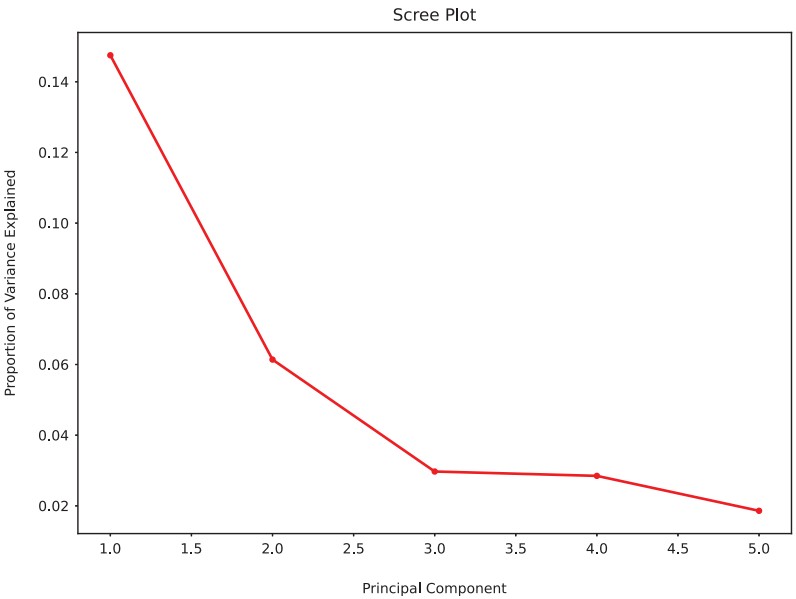

**Figure 12 Principal component analysis (augmented data) summary scree plot.**

where the optimal number of PCs was corresponding to the lowest Root Mean Square Error of Cross Validation (RMSECV) value (Fig. 14). For cross-validation, the testing data were accounted (ten folds). To build the supervised model, 70% of the data was chosen to derive a training data set and the remaining 30% of the data were used for testing the model. This corresponded to 45 training set samples and 23 testing set samples in total for the raw standardised dataset and 50,025 training and 21,443 test samples for the augmented dataset. The lowest value of RMSE corresponded to the component number, that was best suited to describe the highest variable. The number of PLS factors or components for Glucose, for instance, was four (Fig. 14C) for the standardised data set and 10 for the augmented dataset. The different sugars required different numbers of components as shown in Table 2.

The main practical difference between PCR and PLSR is that PCR often needs more components than PLSR to achieve the same prediction error (*Mevik, 2007*; *Mevik & Wehrens, 2015*). On this data set, PCR for Glucose would need seven components (Fig. 14A) to achieve the same Root Mean Square Error of Prediction (RMSEP) for the standardised dataset and 14 for the augmented dataset.

## Artificial neural networks

An ANN is a supervised mathematical model inspired by biological systems. It simulates the way mammalian brains work in processing information and learning from data. The ANN consists of a number of layers that perform different functions, such as multiplying weights, randomly dropping out nodes and performing non-linear activations. Training is performed iteratively, so that as it progresses, the ANN learns by comparing the ground truth to the predictions, and back propagating the losses to adjust the

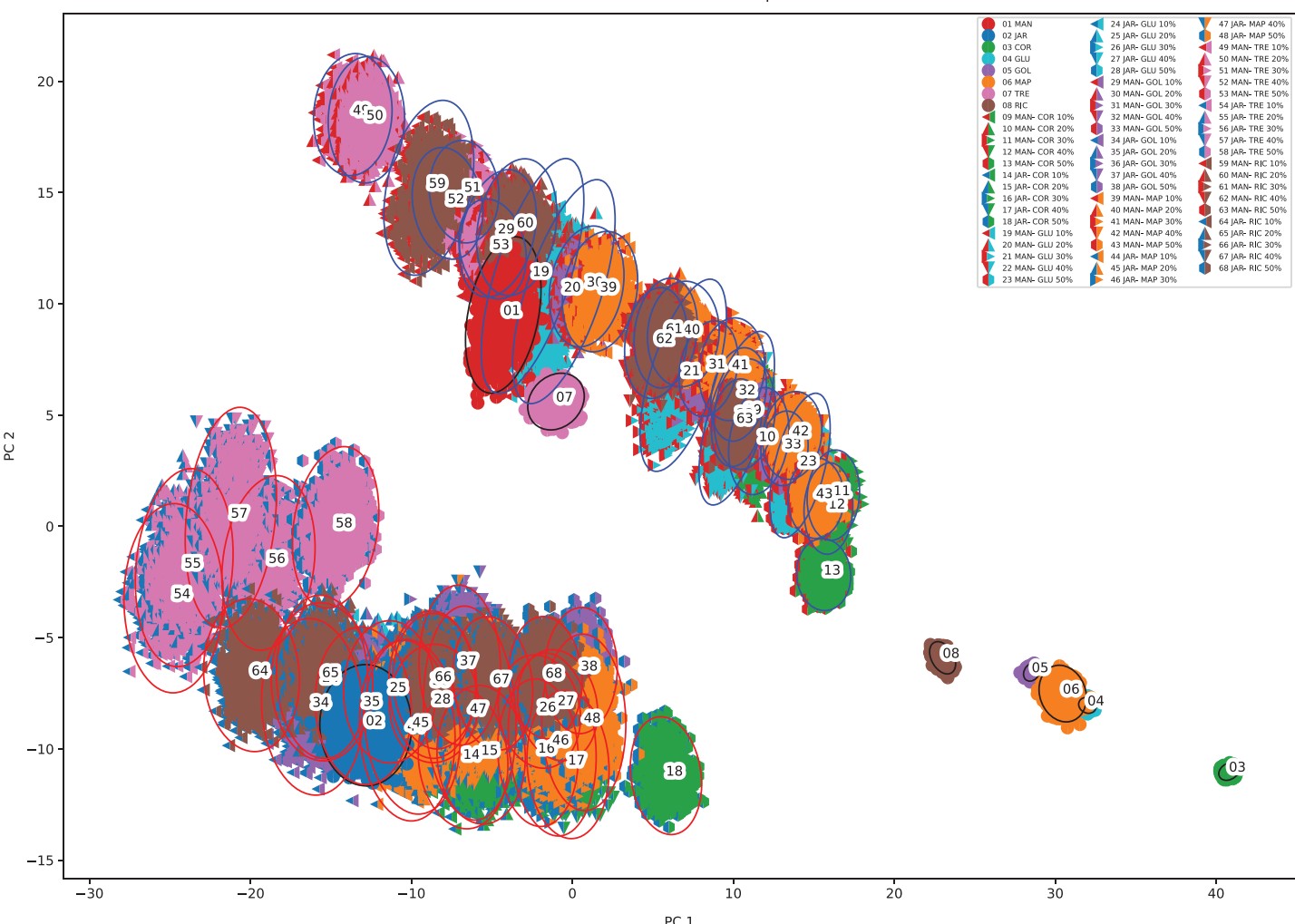

**Figure 13 PCA biplot (augmented data) with clustering.**

**Table 2 Comparison of number of components and RMSE and Rsquare for PCR and PLS method for standardised and augmented dataset.**

| | Standardised dataset | | Augmented dataset | | Standardised dataset | | | | Augmented dataset | | | |
|---|---|---|---|---|---|---|---|---|---|---|---|---|
| | No. of components | | No. of components | | PCR method | | PLS method | | PCR method | | PLS method | |
| | PCR | PLS | PCR | PLS | RMSE | Rsquare | RMSE | Rsquare | RMSE | Rsquare | RMSE | Rsquare |
| Glucose | 7 | 4 | 14 | 10 | 0.7063 | 0.6504 | 0.6328 | 0.7164 | 93.7469 | 0.7553 | 90.7861 | 0.7707 |
| Fructose | 6 | 7 | 12 | 13 | 0.5484 | 0.6629 | 0.5799 | 0.5810 | 147.623 | 0.7828 | 128.596 | 0.8352 |
| Maltose | 1 | 7 | 14 | 14 | 1.0016 | 0.0012 | 0.6541 | 0.2248 | 86.3064 | 0.8537 | 80.254 | 0.8735 |
| Sucrose | 10 | 7 | 15 | 14 | 0.7484 | 0.8605 | 0.6639 | 0.8960 | 84.4529 | 0.8997 | 74.1253 | 0.9227 |

weights of the varies layers. Upon completion the model will have established (linear or non-linear) relationships between input and output data (*Agatonovic-Kustrin & Beresford, 2000*) through the various layers.

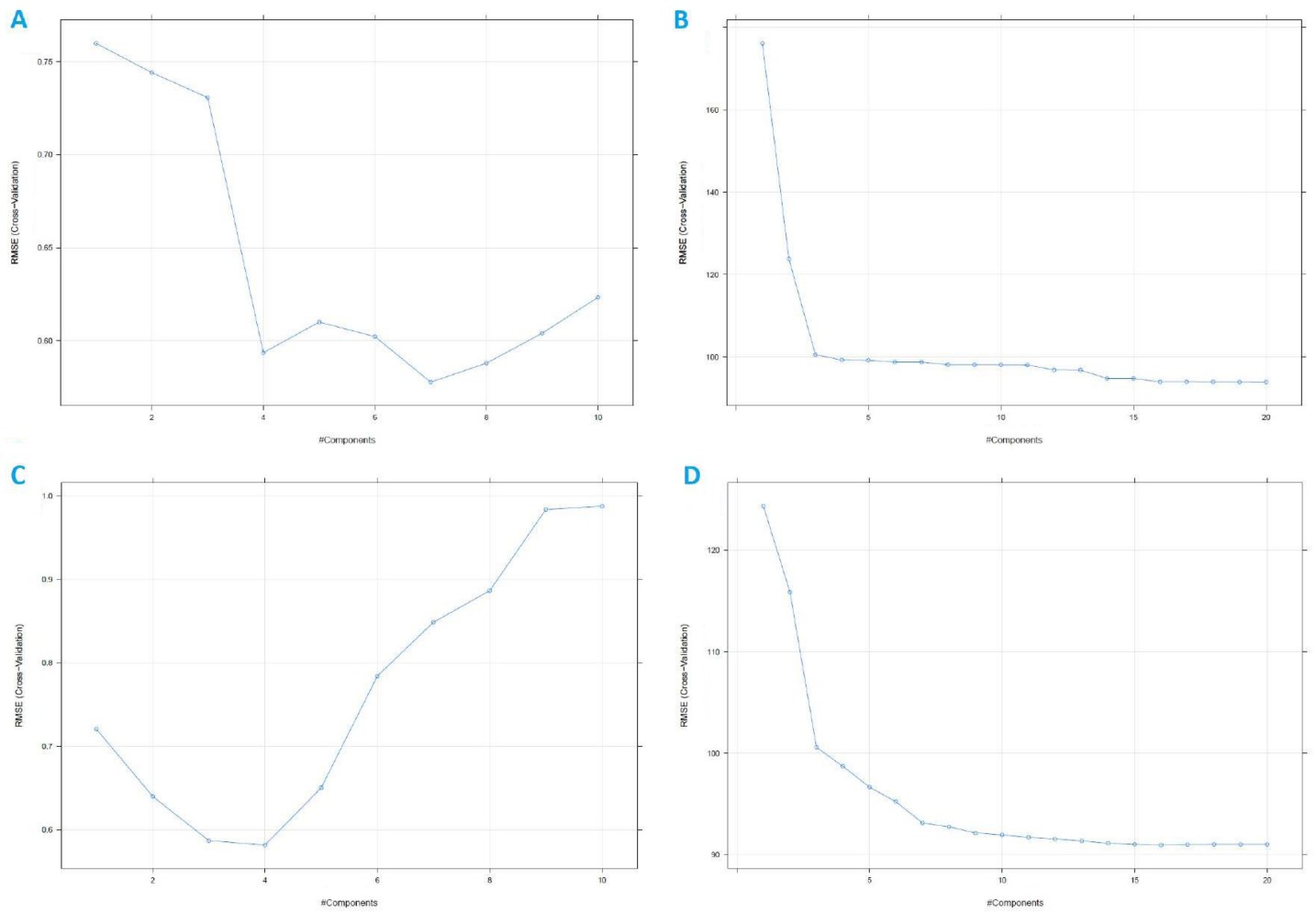

**Figure 14 Partial least squares (PLS) regression RMSE cross-validation *vs* component (A) using PCR method on standardized dataset, (B) using PCR method on augmented dataset, (C) using PLS method on standardized dataset, and (D) using PLS method on augmented dataset (for Fructose, Maltose and Sucrose; see Materials–S3).**

**Table 3 Comparison of accuracy.**

| Fold | Sugars (%) | Organics extracts (%) | Sugars and organics extracts combined (%) |
|------|-----------|----------------------|-------------------------------------------|
| 1 | 92.80 | 94.22 | 99.52 |
| 2 | 93.80 | 95.04 | 99.54 |
| 3 | 93.05 | 94.51 | 99.69 |
| 4 | 92.99 | 94.56 | 99.57 |
| 5 | 93.44 | 94.53 | 99.55 |
| 6 | 93.54 | 94.39 | 99.64 |
| Mean | 93.27 | 94.54 | 99.59 |

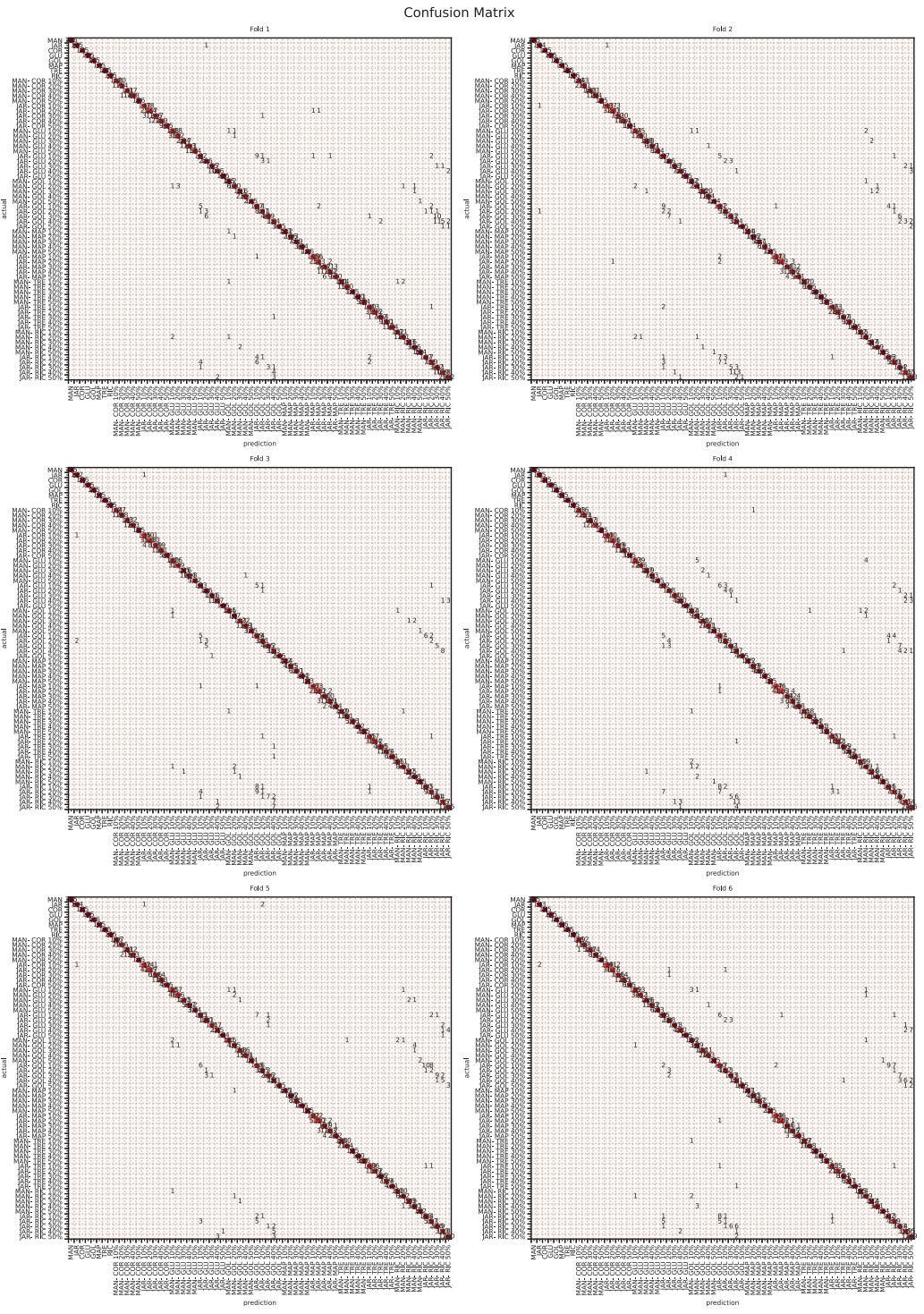

**Figure 15 Confusion matrix for organic extracts Rf *vs* intensity values.**

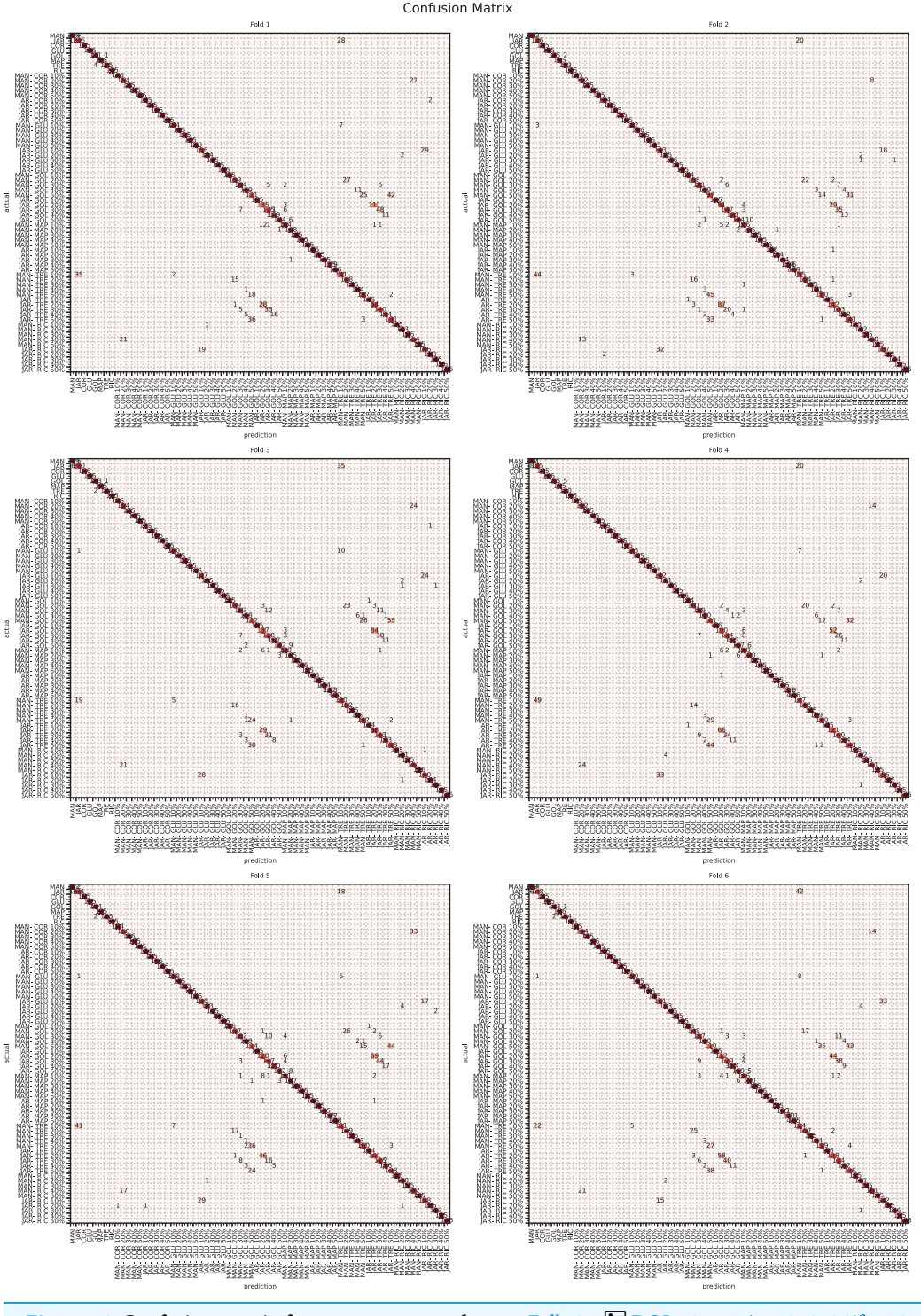

**Figure 16 Confusion matrix for sugar content values.**
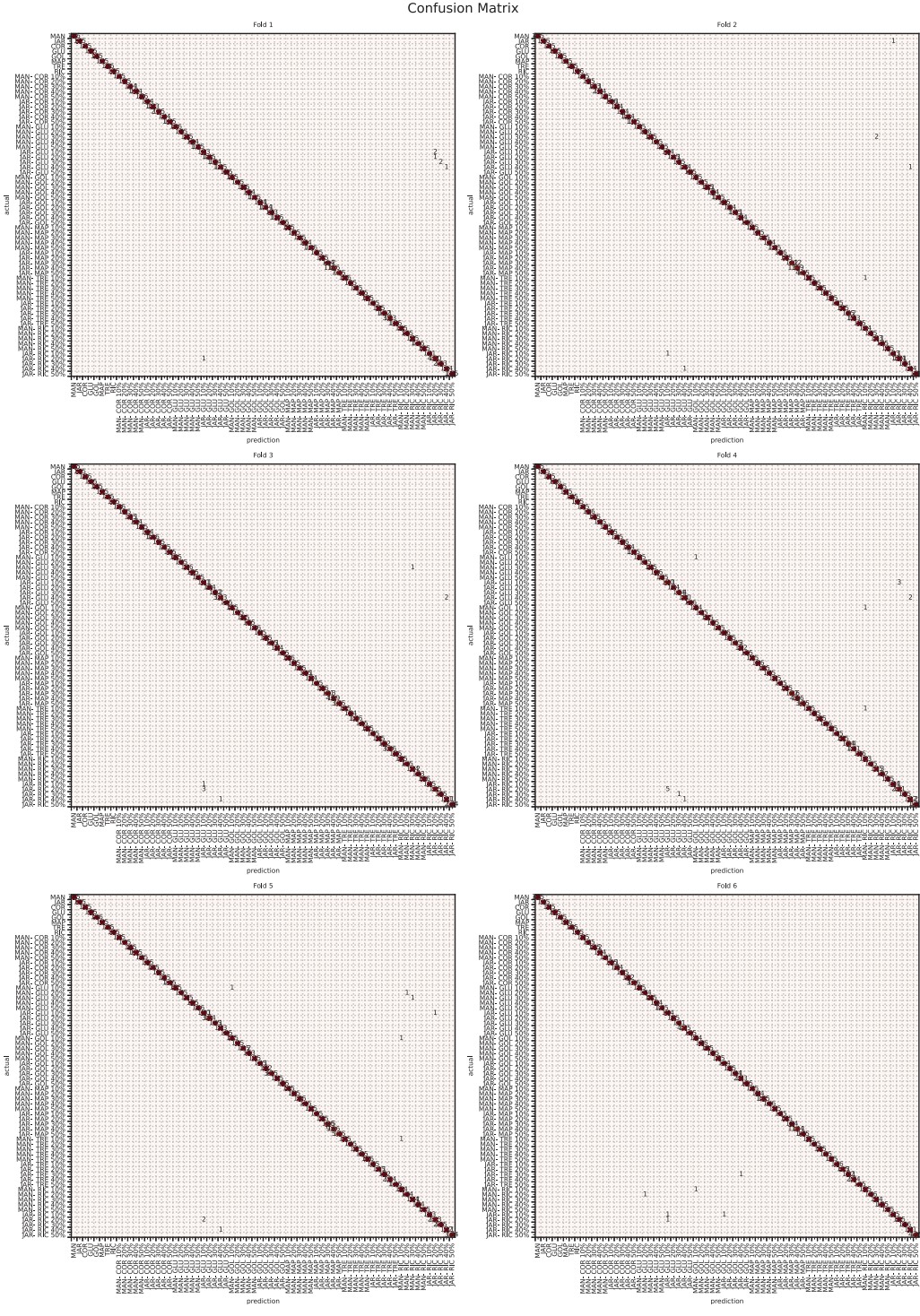

**Figure 17 Confusion matrix for sugar content and organic extracts Rf *vs* intensity values learnt together.**

The augmented data was split into six groups to perform K-fold learning with six folds. For each of the six runs a different fold of the data was retained for testing as discussed in "Chemometric validation". Table 3 shows the final accuracy for each fold for sugars only, organic extracts only and sugars and organic extract combined. Individually the sugars and organic extracts are 93.27% and 94.54% accurate. When combined they reach 99.59% accuracy. The confusion matrixes for the Rf are given in Figs. 15–17. These show that main areas of confusion are the adulterants. A small number of MAN-TRE 10% are misclassified as either JAR or MAN when working with the sugars alone. The organic extracts and sugars plus organic extracts only misclassify the adulterants.

## CONCLUSION

HPTLC derived organic extract profiles allow a reliable and reproducible authentication of a honey's floral source (*Locher, Neumann & Sostaric, 2017*; *Locher et al., 2018*; *Stanek & Jasicka-Misiak, 2018*; *Stanek, Kafarski & Jasicka-Misiak, 2019*), but any accidental or deliberate post-harvest adulteration with sugar syrups cannot be detected with this method alone. It requires an additional layer of analysis, the detection and quantification of simple sugars (glucose, fructose, sucrose, maltose) present in the sample. As has been demonstrated in this study, on the basis of both, HPTLC-derived honey organic extract and sugar profiles, multivariate data analysis allows for a definitive discrimination between pure and sugar syrup adulterated honeys. Cluster Analysis and Principal Component Analysis can easily cluster pure honeys, adulterated honeys and syrup adulterants into separate groups. The use of Partial Least Squares regression and the Artificial Neural Network model can successfully predict the outcome and correctly identify adulterated honeys and the percentage of adulteration based on the available data set of pure honeys and adulterated samples. Specifically, it was found that predictions based on the analysis of the samples' sugar profiles and the samples' organic extract profiles were 93.27% and 94.54% accurate respectively. When the combined data sets (sugar and organic extract profiles) were taken into account, the predictive capacity of ANN exceed 99% accuracy even for samples with post-harvest sugar adulterations of as low as 10% (w/w). This novel approach of combining HPTLC derived organic extract and sugar profile data and subjecting them to multivariate data analysis might therefore offer a powerful tool for the detection of post-harvest sugar syrup adulterants in the quality control of honeys.

## ACKNOWLEDGEMENTS

We also thank Chromatech Scientific for their technical support of this project.

### Funding

This research was funded by the Cooperative Research Centre for Honey Bee Products (CRC HBP). The funders had no role in study design, data collection and analysis, decision to publish, or preparation of the manuscript.

## Grant Disclosures

The following grant information was disclosed by the authors:
Cooperative Research Centre for Honey Bee Products (CRC HBP).

## Competing Interests

The authors declare that they have no competing interests. Md Khairul Islam and Cornelia Locher are affiliated with the CRC for Honey Bee Products, which is an umbrella organisation which brings together industry and government organisations with industry. Md Khairul Islam received a scholarship from the CRC for his PhD research and Cornelia Locher is a full time employee of the University of Western Australia and has therefore no financial dependence on the CRC.

## Author Contributions

- Md Khairul Islam conceived and designed the experiments, performed the experiments, analyzed the data, prepared figures and/or tables, authored or reviewed drafts of the paper, and approved the final draft.
- Kevin Vinsen conceived and designed the experiments, performed the experiments, analyzed the data, prepared figures and/or tables, authored or reviewed drafts of the paper, and approved the final draft.
- Tomislav Sostaric performed the experiments, analyzed the data, authored or reviewed drafts of the paper, and approved the final draft.
- Lee Yong Lim conceived and designed the experiments, authored or reviewed drafts of the paper, supervision, and approved the final draft.
- Cornelia Locher conceived and designed the experiments, authored or reviewed drafts of the paper, supervision, and approved the final draft.

## Data Availability

The raw data are available in a Supplementary File.

## Supplemental Information

Supplemental information for this article can be found online at http://dx.doi.org/10.7717/peerj.12186#supplemental-information.

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
