# Peer review of "Detection of syrup adulterants in manuka and jarrah honey using HPTLC-multivariate data analysis"

_PeerJ, doi:10.7717/peerj.12186_

## Round 0.1 · original submission · Major Revisions

As you can see, the reviewers have provided extensive comments on the manuscript (some in the attached file). Please address all these comments carefully.

Reviewer 1 ·

Basic reporting

The manuscript (62039) reports on the application of HPTLC to honey adulterants.
Recently, HPTLC has been applied to chemical profiling of natural products including honey. The authors combined various chemometrics methods to HPTLC for the purpose.
Despite the efforts, there is lack of novelty in the manuscript. There are already quite number of works of HPTLC to honey samples, both for sugars and minor floral metabolites. Especially, to prove the feasibility of the methods, more number of honey samples are required.
The application of HPTLC did not show any clear advantages over other methods. For examples, compared with GC analysis, the advantages are not clear, though the authors said GC is expensive (actually, the cost of instruments are not that different).

Experimental design

The target of metabolites are in this study, sugars and organic phase metabolites. The analysis of the honey samples adulterated with sugars by HPTLC is not that new. The analysis of non polar metabolites are already well known. Even in the study, there is not detailed work for non-polar part. Also, in non-polar metabolites of honey, there are many terpenoids, non-phenolic metabolites, but in the manuscript there is no data interpretation about there terpenoidal metabolites.

Validity of the findings

Data obtained in the study is validated but there must be more honey samples for validating feasibility.

Additional comments

It is interesting to apply HPTLC to adulterated honey samples. However, targeting on sugar detection is already well known. There is not much novelty in the method.

Reviewer 2 ·

Basic reporting

The text is not well structured,it also contains numerous small paragrahps that have to be merged. The background and justification of the study is not enough. For example, if there are several methods for honey autentication, what are the real avantages of implementing HPTLC? Moreover, even if there is a raw data file, this have to be properly presented (please see specific comments). Finally, results are not described enough and the discussion is more focused on explining statistical analysis procedures (which are not developed by the authors) rather than focus on discusing their own results.

Experimental design

The research question is well stablished, however, not well expressed (please see specific comments). There are important missing information on methods which can change results obtained when trying to replicate the study.

Validity of the findings

There was just one replicate for autentic honeys and the syrups, and results of each multivariate data anlysis is not properly interpretated.

Additional comments

The topic presented by the authors is original and used a reemerging analytical technique in combination with multivariate data analysis. Moreover, the samples analyzed (honeys) are quite interesting. Nonetheless, there are several format and content issues to be corrected. Please, get much better quality figures.

Annotated reviews are not available for download in order to protect the identity of reviewers who chose to remain anonymous.

---

## Round 0.2 · accepted · Accept

The authors have responded to the reviewers' comments.